# Joint Detection and Reconstruction of Weak Spectral Lines under Non-Gaussian Impulsive Noise with Deep Learning

Zhen Li [1,2,3], Junyuan Guo [1,2,3] and Xiaohan Wang [1,2,3,*]

1 Acoustic Science and Technology Laboratory, Harbin Engineering University, Harbin 150001, China; greenhandli@hrbeu.edu.cn (Z.L.); guojunyuan@hrbeu.edu.cn (J.G.)
2 Key Laboratory of Marine Information Acquisition and Security (Harbin Engineering University), Ministry of Industry and Information Technology, Harbin 150001, China
3 College of Underwater Acoustic Engineering, Harbin Engineering University, Harbin 150001, China
* Correspondence: wangxiaohan@hrbeu.edu.cn

**Abstract:** Non-Gaussian impulsive noise in marine environments strongly influences the detection of weak spectral lines. However, existing detection algorithms based on the Gaussian noise model are futile under non-Gaussian impulsive noise. Therefore, a deep-learning method called AINP+LR-DRNet is proposed for joint detection and the reconstruction of weak spectral lines. First, non-Gaussian impulsive noise suppression was performed by an impulsive noise preprocessor (AINP). Second, a special detection and reconstruction network (DRNet) was proposed. An end-to-end training application learns to detect and reconstruct weak spectral lines by adding into an adaptive weighted loss function based on dual classification. Finally, a spectral line-detection algorithm based on DRNet (LR-DRNet) was proposed to improve the detection performance. The simulation indicated that the proposed AINP+LR-DRNet can detect and reconstruct weak spectral line features under non-Gaussian impulsive noise, even for a mixed signal-to-noise ratio as low as −26 dB. The performance of the proposed method was validated using experimental data. The proposed AINP+LR-DRNet detects and reconstructs spectral lines under strong background noise and interference with better reliability than other algorithms.

**Keywords:** non-Gaussian impulsive noises; detection and reconstruction of weak spectral lines; deep learning

## 1. Introduction

The single-frequency detection of underwater radiation noise with abundant single-frequency components is crucial for detecting quiet targets [1]. The time-frequency analysis is projected on the time and frequency planes to form a three-dimensional stereogram (lofargram). It presents the abundant features of underwater radiation noise [2]. Therefore, the lofargram is regularly employed to analyze its features for passive sonar signals. However, for low signal-to-noise ratios (SNRs), frequency fluctuations caused by a moving target [3] and a high amount of background noise may weaken spectral-line detection.

The detection of weak spectral lines using a lofargram has long been an attractive research topic. Image-processing methods, neural networks, and statistical models are applied to detect weak spectral lines in a lofargram. Image-processing and neural-network methods obtain spectral-line traces from complex image semantic features; however, their performance is usually unsatisfactory for low SNRs [4–7]. Some deep-learning methods [8–11] achieve good line-spectrum estimation, but the SNR requirement is relatively high. To overcome this limitation, deepLofargram was proposed to recover invisible and irregularly fluctuating frequency lines at low SNRs [12]. Furthermore, in lofargrams, when the weak spectral lines are far beyond the perceptual range of human vision, this is referred to as low SNR. A statistical model such as the hidden Markov model (HMM) can track the optimal spectral-line trajectory from multi-frame power-spectrum data [13,14]. Most of

the aforementioned studies were applied to marine ambient noise following a Gaussian distribution. In particular, marine ambient noise presents strong impulsive characteristics owing to the superposition of seawater thermal noise, hydrodynamic noise, under-ice noise, biological noise, and other noises [15]. Existing underwater-acoustic-signal-processing methods may be invalidated under such non-Gaussian impulsive noise. To overcome this problem, several studies [16–18] have performed statistical analyses and models of non-Gaussian marine ambient noise. It was found that the generation and propagation of underwater impulsive noise are in accordance with the "heavy tail" statistical characteristics of the symmetric $\alpha$-stable (S$\alpha$S) distribution [19]. Furthermore, SNR and mixed signal-to-noise ratio (MSNR) have been used to characterize the energies of Gaussian noise and non-Gaussian impulsive noise [19]. Various preprocessors have been proposed to suppress non-Gaussian impulsive noise, which can be described by the S$\alpha$S distribution, including the standard median filter (SMF) [20–22] and the memoryless analog nonlinear preprocessor (MANP) [23]. Nevertheless, weak spectral-line detection is unreliable at low MSNRs.

In recent years, with the introduction and development of deep convolutional structures such as UNet [24], SegNet [25], and LinkNet [26], image-semantic-segmentation technology based on deep learning has developed rapidly. In deep-learning semantic segmentation, the semantic features in an image are captured by finding semantic correlations between pixel points from global or local contextual information. In passive sonar-signal processing, weak spectral lines have time-frequency correlations, making them relatively continuous in a lofargram, even though they cannot be observed. Therefore, we argue that when combined with a preprocessor and a deep convolution structure, a lofargram would be able to handle the detection and reconstruction of weak spectral lines under non-Gaussian impulsive noise. Moreover, by "reconstruction," we mean that potential spectral-line features are recovered to output a lofargram with significant spectral lines.

In this study, we propose a novel method, called AINP+LR-DRNet, which is suitable for the detection and reconstruction of weak spectral lines under non-Gaussian impulsive noise. The spectral-line detection-and-reconstruction problem is redefined as a binary classification problem. First, an impulsive noise preprocessor (AINP) was applied to suppress the non-Gaussian impulsive noise. Second, a specially constructed DRNet was built to detect and reconstruct weak spectral lines. Third, a dual classification adaptive weighted loss was applied to obtain the optimal DRNet during training iterations. Fourth, the detection performance was further improved by the proposed LR-DRNet algorithm. Finally, we validated the ability of the proposed method to detect and reconstruct weak spectral lines under non-Gaussian-impulse noise using simulated and measured data sets.

## 2. Proposed Framework and Training

Considering deep learning (DL) techniques, we formulate the spectral-line detection-and-reconstruction problem in a lofargram as a binary classification problem. Thus, binary hypothesis testing can be performed, which is defined as follows:

$$
\begin{aligned}
H_1 : & \sum_{i,j}^{i=T,j=F} [s(t_i, f_j) + u(t_i, f_j)] \\
H_0 : & \sum_{i,j}^{i=T,j=F} u(t_i, f_j)
\end{aligned}, \tag{1}
$$

where $H_1$ and $H_0$ indicate the presence of spectral-line pixels and noise pixels in a lofargram, respectively. $\sum_{i,j}^{i=T,j=F} s(t_i, f_j)$ describes the set of spectral-line pixels, and $\sum_{i,j}^{i=T,j=F} u(t_i, f_j)$ describes the set of noise pixels.

Thus, the spectral-line detection-and-reconstruction framework are proposed to solve Equation (1). As shown in Figure 1, the proposed framework, with the sampling, detection, and reconstruction algorithm, is illustrated. In the sampling stage, the passive SONAR

system collects the acoustic signals and noises. The received data are preprocessed by AINP to construct the dataset. Subsequently, a specially designed LR-DRNet is pre-trained to obtain the optimal model parameters in offline training by adding into an adaptive weighted loss function based on dual classification. The well-trained LR-DRNet is utilized to fine-tune the parameters to detect and reconstruct the measured unlabeled samples in online detection and reconstruction. More details are described below:

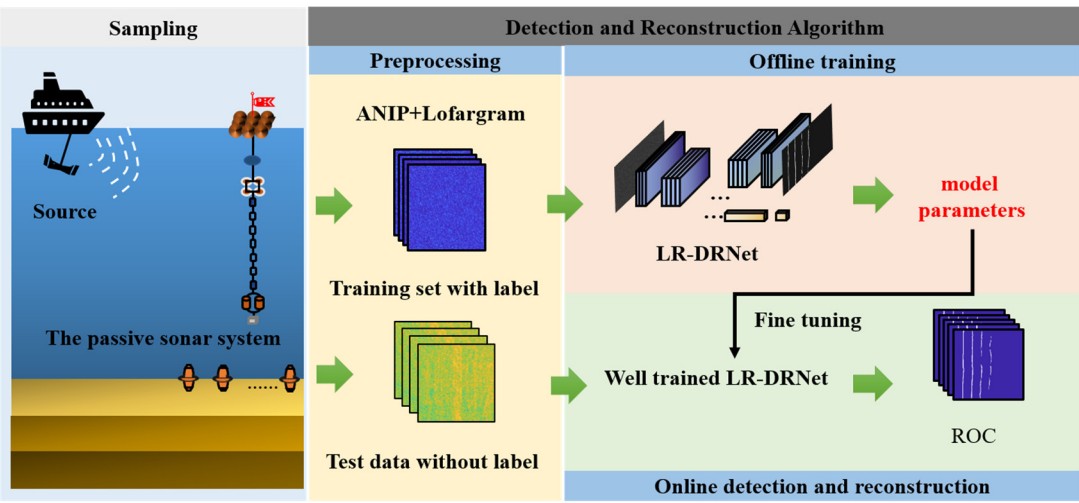

**Figure 1.** Proposed spectral-line detection-and-reconstruction framework.

### 2.1. Detection-and-Reconstruction Algorithm

#### 2.1.1. Data Preprocessing

The heavy impulsive noise causes broadband interference in a lofargram. Therefore, appropriate preprocessing is required. In this study, the AINP method [24] is used to nonlinearly suppress the abnormal amplitude in the input signal $s(t)$, which is more prominent than the amplitude threshold $\theta(t)$. The influence function for the AINP is as follows:

$$e(t) = s(t) \begin{cases} 1 \, , & |s(t)| \leq \theta(t) \\ (\frac{\theta(t)}{|s(t)|})^2, & |s(t)| > \theta(t) \end{cases}, \tag{2}$$

where $\theta(t)$ can be obtained from Equation (3)

$$\theta(t) = (1 + 2\theta_0)Q_2(t). \tag{3}$$

In Equation (3), $Q_2(t)$ represents the second quartile of the absolute value of the input signal $|g(t)|$, and $\theta_0$ is a coefficient, which is set to 1.5, as in [23].

#### 2.1.2. DRNet Structure

The proposed DRNet is derived from LinkNet [26], including the shared encoder, detection decoder, and reconstruction decoder. One part of the shared encoder, illustrated in Figure 2a, is stacked with a series of residual convolution structures to extract spectral-line features. For spectral-line detection, the detection decoder is added to output result of spectral-line detection (for $H_1$ and $H_0$, respectively). As illustrated in Figure 2b, plugging into the squeeze-and-excitation (SE) blocks, the reconstruction decoder can perform channel enhancement by obtaining the importance of each channel through "squeeze" and "excitation" operations [27]. The reconstruction decoder outputs spectral line-reconstruction results through FinalConv, as shown in Figure 2c. Moreover, a reconstruction decoder is enabled when the detection decoder announces that $H_1$ holds during the online detection-and-reconstruction stage. The complete network structure is depicted in Figure 3. As shown in Figure 3, the proposed DRNet involves multi-task learning (MTL). For MTL, the model

relies on the relative weighting between each task's loss, and manually adjusting these weights is difficult and time-consuming. Hence, inspired by [28], the adaptive weighted dual loss function is considered by modeling reconstruction-and-detection-task uncertainty. The details are described in Section 2.1.3.

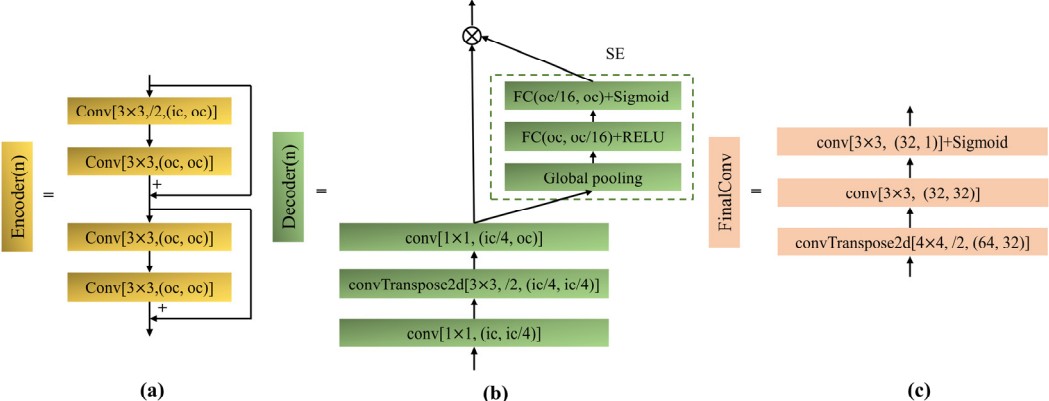

**Figure 2.** Structural diagram of each part in DRNet. (**a**) Structure of convolutional modules in Encoder (n). (**b**) Structure of the decoding layer. (**c**) Structure of the FinalConv layer.

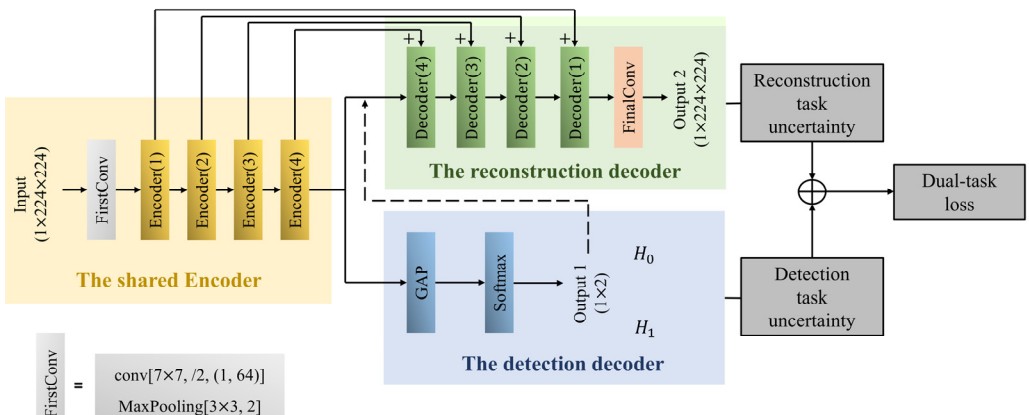

**Figure 3.** Architecture of DRNet.

### 2.1.3. Adaptive Weighted Loss Function Based on Dual Classification

For MTL, the loss function is weighted for each task loss. Thereafter, the MTL loss can be expressed as:

$$L = \sum_{i=1}^{I} w_i L_i, \tag{4}$$

where $w_i$ and $L_i$ denote the weight and loss of the $i$-th task, respectively. The $I$ indicates the number of tasks. In this study, there are spectral-line detection and reconstruction tasks with different loss scales. A basic approach to overcoming the large loss difference between the detection and reconstruction tasks involves the model adaptively adjusting the weights $w_i$ according to uncertainty of each task.

As the spectral-line detection-and-reconstruction problem is treated as a binary classification task, an adaptive weighted loss function based on a dual classification is used to train the model. Following the derivation in [28], when $f^W(x)$ is a sufficient statistic, the following multi-mask likelihood is expressed as

$$p(y_d, y_r | f^W(x)) = p(y_d | f^W(x)) p(y_r | f^W(x)), \tag{5}$$

where $f^W(x)$ represents the detection-and-reconstruction-prediction results of model with parameters $W$ on input $x$. $y_d$ and $y_r$ are the ground-truth labels for detection and reconstruction tasks.

Under random noise, the log-likelihood of the detection and reconstruction task is output through a Softmax function, which can be expressed as

$$\log p(y_d|f^W(x)) = \log(\text{Softmax}(\frac{1}{\sigma_d^2}f_d^W(x))) = \frac{1}{\sigma_d^2}f_d^W(x) - \log \sum_{c_1} \exp(\frac{1}{\sigma_d^2}f_{c_1}^W(x)), \quad (6)$$

$$\log p(y_r|f^W(x)) = \log(\text{Softmax}(\frac{1}{\sigma_r^2}f_r^W(x))) = \frac{1}{\sigma_r^2}f_r^W(x) - \log \sum_{c_2} \exp(\frac{1}{\sigma_r^2}f_{c_2}^W(x)), \quad (7)$$

where $c_1$ and $c_2$ denote the categories of the detection and reconstruction tasks, respectively. The $\sigma_d^2$ and $\sigma_r^2$ denote the observation-noise parameters of the model for the detection and reconstruction tasks, respectively.

When the Softmax likelihood is modeled for the detection and reconstruction, the joint loss $L(W, \sigma_d, \sigma_r)$ is

$$
\begin{aligned}
L(W, \sigma_d, \sigma_r) &= -\log p(y_d, y_r|f^W(x)) \\
&= -\log[\text{Softmax}(\frac{1}{\sigma_d^2}f_d^W(x)) \cdot \text{Softmax}(\frac{1}{\sigma_r^2}f_r^W(x))] \\
&= \frac{1}{\sigma_d^2}f_d^W(x) - \log \sum_{c_1} \exp(\frac{1}{\sigma_d^2}f_{c_1}^W(x)) + \frac{1}{\sigma_r^2}f_r^W(x) - \log \sum_{c_2} \exp(\frac{1}{\sigma_r^2}f_{c_2}^W(x)) \\
&= \frac{1}{\sigma_d^2}[f_d^W(x) - \log \sum_{c_1} \exp(\frac{1}{\sigma_d^2}f_{c_1}^W(x))] + \frac{1}{\sigma_r^2}[f_r^W(x) - \log \sum_{c_2} \exp(\frac{1}{\sigma_r^2}f_{c_2}^W(x))] \\
&+ \log \sum_{c_1} \exp(\frac{1}{\sigma_d^2}f_{c_1}^W(x))/(\sum_{c_1} \exp(f_{c_1}^W(x)))^{\frac{1}{\sigma_d^2}} + \log \sum_{c_2} \exp(\frac{1}{\sigma_r^2}f_{c_2}^W(x))/(\sum_{c_2} \exp(f_{c_2}^W(x)))^{\frac{1}{\sigma_r^2}} \\
&\approx \frac{1}{\sigma_d^2}[-\log(\text{Softmax}(y_d, f_d^W(x)))] + \frac{1}{\sigma_r^2}[-\log(\text{Softmax}(y_r, f_r^W(x)))] + \log \sigma_d \sigma_r \\
&= \frac{1}{\sigma_d^2}L_d(W) + \frac{1}{\sigma_r^2}L_r(W) + \log \sigma_d \sigma_r
\end{aligned} \quad (8)
$$

where $f_d^W(x)$ and $f_r^W(x)$ represent the outputs of detection and reconstruction in $f_{c_1}^W(x)$ and $f_{c_2}^W(x)$, respectively. Equation (8) can be applied for approximation, as follows:

$$\sum_{c_1} \exp(\frac{1}{\sigma_d^2}f_{c_1}^W(x))/\sum_{c_1} \exp(f_{c_1}^W(x))^{\frac{1}{\sigma_d^2}} \approx \sigma_d, \sum_{c_2} \exp(\frac{1}{\sigma_r^2}f_{c_2}^W(x))/\sum_{c_2} \exp(f_{c_2}^W(x))^{\frac{1}{\sigma_r^2}} \approx \sigma_r, \quad (9)$$

where Equation (9) becomes equal when $\sigma_d, \sigma_r \to 1$.

Referring to the suggestion in [28], we set $s_d = \log \sigma_d^2$, $s_e = \log \sigma_e^2$. Accordingly, $L(W, \sigma_d, \sigma_r)$ can be rewritten as

$$L(W, \sigma_d, \sigma_r) = 2\exp(-s_d)L_d(W) + 2\exp(-s_r)L_r(W) + s_d + s_e. \quad (10)$$

For the detection and reconstruction loss functions $L_d(W)$ and $L_r(W)$, we adopt the two-class cross-entropy and the class-balanced cross-entropy loss functions in [12], as follows:

$$L_d(W) = -\sum_{i=1}^{N_1} h \log p + (1-h)\log(1-p), \quad (11)$$

where $N_1$ indicates the number of samples in the batch size. Moreover, $h \in \{0, 1\}$ represents the $H_0$ and $H_1$ hypotheses, and $p$ indicates the probability of the output sample class when using a Softmax function.

$$L_r(W) = \sum_{i=1}^{N_2} \lambda[\sum_{f,t \in G_+} \log p_{f,t} + (1-\lambda)\sum_{f,t \in G_-} \log(1-p_{f,t})], \quad (12)$$

where $\lambda = |G_-|/|G|$ and $1 - \lambda = |G_+|/|G|$. The $|G_+|$ and $|G_-|$ represent the spectral-line and noise ground-truth label sets, respectively. The $p_{f,t}$ indicates the predicted value of the $H_1$ samples at the $(f, t)$ position by a sigmoid function.

According to Equations (10)–(12), the joint-loss form of the multi-task can be obtained. Simultaneously, two weight parameters, $\sigma_d$ and $\sigma_r$, are adaptively adjusted during the training process. Thus, the purpose of adaptive loss weighting is achieved.

### 2.1.4. DRNet-Based Spectral-Line-Detection Algorithm

Inspired by the application of the CNN-based spectrum sensing algorithm [29] in narrowband spectrum sensing, which provides a path for detecting spectral lines in a lofargram, a LR-DRNet algorithm is proposed by considering only a single receiver hydrophone. In the proposed algorithm, we use DRNet for offline training and adopt a threshold-based mechanism for online detection.

### Offline Training

In offline training, the dataset of the lofargram is constructed under $H_0$ and $H_1$ after applying AINP and labeled as follows:

$$(O_l, Z) = \left\{ (l^{(1)}, z^{(1)}), (l^{(2)}, z^{(2)}), ..., (l^{(M)}, z^{(M)}) \right\}, \tag{13}$$

where $O_l$ denotes the set of lofargrams $l$, and $Z$ is its label. The $(l^{(m)}, z^{(m)})$ represents the m-th sample in the training set.

For the test statistic, the proposed LR-DRNet can extract weak spectral-line features in a lofargram. The output node of the detection decoder was set to 2 by converting spectral line detection into an image binary classification. After a series of convolutional layers, pooling layers, and activation functions, the probability that the lofargram belongs to $H_0$ or $H_1$ can be obtained. For the detection task, Equation (1) can be rewritten as follows:

$$
\begin{aligned}
H_1 &: P(z^{(m)} = 1 \big| l^{(m)}; \vartheta) = h_{\vartheta|H_1}(l^{(m)}) \\
H_0 &: P(z^{(m)} = 0 \big| l^{(m)}; \vartheta) = h_{\vartheta|H_0}(l^{(m)})
\end{aligned}' \tag{14}
$$

where $h_\vartheta(\cdot)$ represents a nonlinear expression of the model with parameters $\vartheta$. After a Softmax function, the network's output layer has:

$$h_{\vartheta|H_1}(l^{(m)}) + h_{\vartheta|H_0}(l^{(m)}) = 1. \tag{15}$$

Next, Equation (11), as a training-error loss function, can be rewritten as:

$$J_{LR-DRNet}(\vartheta) = -\frac{1}{K} \sum_{k=1}^{K} \left\{ z^{(m)} \log h_{\vartheta|H_1}(l^{(m)}) + (1 - z^{(m)}) \log h_{\vartheta|H_0}(l^{(m)}) \right\}. \tag{16}$$

Training LR-DRNet minimizes the error loss in Equation (16) and maximizes the posterior probability of the parameter set $\vartheta$. The optimal parameter set $\widehat{\vartheta}$ can be obtained as follows:

$$\widehat{\vartheta} = \mathrm{argmax} P(Z|L; \vartheta). \tag{17}$$

Based on the loss function in Equation (10), the backpropagation algorithm is employed to gradually update the parameters of LR-DRNet. Hence, the well-trained LR-DRNet can be illustrated as follows:

$$h_{\widehat{\vartheta}|H_i}(l^{(m)}) = \begin{cases} h_{\widehat{\vartheta}|H_1}(l^{(m)}) \\ h_{\widehat{\vartheta}|H_0}(l^{(m)}) \end{cases}. \tag{18}$$

Considering the Bayesian and Neyman–Pearson (NP) criterion, and assuming that $P(H_0) = P(H_1)$, the test statistics under the proposed LR-DRNet can be acquired as:

$$\Lambda_{LR-DRNet} = \frac{h_{\widehat{\vartheta}|H_1}(l^{(m)})}{h_{\widehat{\vartheta}|H_0}(l^{(m)})} \gtrless \eta, \tag{19}$$

where $\eta$ denotes the detection threshold. The presence or absence of spectral lines in the lofargram can be adjudicated by comparing the test statistic and detection threshold.

Next, the detection threshold should be determined. First, M, noise-sample data sets composed of $H_0$ lofargrams after applying AINP, are constructed. M lofargrams under $H_0$ after applying AINP are costructed.

$$O_n = \left\{ n^{(1)}, n^{(2)}, ..., n^{(M)} \right\}. \tag{20}$$

where $O_n$ denotes the set of lofargrams $n$ under $H_0$.

The probability of detection ($P_D$) and the false-alarm probability ($P_f$) are defined as follows:

$$P_D = P[\Lambda_{LR-DRNet}|H_1 > \eta], \tag{21}$$

$$P_f = P[\Lambda_{LR-DRNet}|H_0 > \eta]. \tag{22}$$

Subsequently, the data set $O_n$ is fed as samples into the pre-trained DRNet and the test statistics of all lofargrams under the $H_0$ hypothesis are obtained.

$$\Lambda_{LR-DRNet}|H_0 = \frac{h_{\widehat{\vartheta}|H_1}(n^{(m)})}{h_{\widehat{\vartheta}|H_0}(n^{(m)})}. \tag{23}$$

By arranging these values in descending order to form a sequence $T(m)$, the detection threshold of the artificially set false-alarm-probability value $P_f$ can be acquired [29].

$$\Lambda_{LR-DRNet}|H_i(\tilde{z}) = \frac{h_{\hat{\gamma}|H_1}(l^{(m)})}{h_{\hat{\gamma}|H_0}(l^{(m)})} \begin{matrix} H_1 \\ \gtrless \\ H_0 \end{matrix} \eta. \tag{24}$$

where $\lfloor \cdot \rfloor$ is the nearest smaller integer. The $\Lambda_{LR-DRNet}|H_0(\lfloor m \rfloor)$ denotes the $m$-th sample value of $T(m)$ in descending order.

Online Detection

According to Equation (24), a detection threshold $\eta$ is set. The unlabeled lofargrams, denoted as $\tilde{z}$, are input into the well-trained LR-DRNet. Subsequently, online detection, based on LR-DRNet, is performed, that is,

$$\Lambda_{LR-DRNet}|H_i(\tilde{z}) = \frac{h_{\hat{\gamma}|H_1}(l^{(m)})}{h_{\hat{\gamma}|H_0}(l^{(m)})} \begin{matrix} H_1 \\ \gtrless \\ H_0 \end{matrix} \eta. \tag{25}$$

When the test statistic is obtained, we can rapidly decide whether there are spectral lines in a lofargram by comparing it to the preset threshold.

### 2.2. Training Process

Training was optimized for the loss function in Equation (10) using the mini-batch gradient of the Adam optimizer [30], and by setting $s_d$ and $s_e$ to log2. The batch size was 128. Xavier weight initialization was performed [31]. As expressed in [12], the network's

loss function is ineffective at converging at low SNRs. Hence, we first pre-trained the model with a learning rate of $10^{-4}$ for lofargrams with MSNR ranging from $-19$ dB to $-22$ dB. The model was then retrained with a learning rate of $10^{-5}$ for lofargrams with MSNR ranging from $-23$ dB to $-26$ dB. The learning rate was not fixed and was adjusted according to the cosine annealing warm restart [32] and gradual warmup [33]. Here, the gradual warmup was up to the 10th epoch, the initial restart epoch was set to 15, and the restart factor was set to 2. To prevent network overfitting and the problem of insufficient data, data augmentation was performed during training using methods such as horizontal and vertical flipping of images, random cropping, and grayscale maps. Both the above training procedures were terminated after approximately 300 epochs.

## 3. Simulation Analysis

This section first introduces the synthesis of the datasets and the network performance evaluation metrics. Subsequently, we illustrate the effectiveness of the proposed method by analyzing the effect of the network structure on performance. Finally, the performances of some existing methods are compared and analyzed through simulations.

### 3.1. Datasets

Non-Gaussian impulsive noise can be described by an $\alpha$-stable distribution, whose characteristic function can be expressed as in [34].

$$\varphi(t) = \exp\{jbt - |\gamma t|^{\alpha}[1 + j\beta\mathrm{sgn}(t)\omega(t,\alpha)]\}, \tag{26}$$

$$\omega(t,\alpha) = \begin{cases} -\tan(\frac{\pi\alpha}{2}), & \alpha \neq 1 \\ (\frac{2}{\pi})\log|t|, & \alpha = 1 \end{cases}, \tag{27}$$

where $0 < \alpha \leq 2$, $-1 \leq \beta \leq 1$, $\gamma > 0$, and $-\infty < b < \infty$. The characteristic exponent $\alpha$ determines the impulse intensity of the distribution; the higher the value of $\alpha$, the lower the intensity. The position parameter $b$ determines the center of the distribution. The dispersion coefficient $\gamma$ measures the sample's degree of deviation by taking values relative to the mean, which is similar to the variance in a Gaussian distribution. The symmetry parameter $\beta$ is used to describe the skewness of the distribution. When $\beta = 0$, the distribution is named the S$\alpha$S distribution.

In this case of $0 < \alpha \leq 2$, only the first order is presented in $\alpha$-stable distributed noise. Therefore, the SNR defined under traditional Gaussian noise is inapplicable. The mixed signal-to-noise ratio (MSNR) is defined as follows:

$$MSNR = 10\log_{10}(\frac{v_s^2}{\gamma}), \tag{28}$$

where $v_s^2$ denotes the signal variance.

The $\alpha$-stable distribution degenerates into a Gaussian distribution when $\alpha = 2$. A conventional SNR measure of the relationship between the signal and noise power can be obtained as follows:

$$SNR = 10\log_{10}(\frac{a^2}{v_n^2}), \tag{29}$$

where $\alpha$ and $v_n^2$ denote the signal amplitude and the noise variance, respectively.

A low-frequency spectral line, radiated by underwater and surface vehicles, under Gaussian/non-Gaussian impulsive noise, is discussed in this study. Owing to the motion of vehicles (variable speed or steering), the spectral lines fluctuate even at low frequencies.

The fluctuating spectral lines can be simulated using a series of sinusoidal signals. The fluctuating spectral lines observed during the k-th time interval are described as:

$$s(t_k) = \sum_{i=1}^{I} a_i(t_k)\sin(2\pi f_i(t_k) + \varphi_k) + n(t_k), k = 0, 1, ..., T-1, \tag{30}$$

where $I$ represents the number of spectral lines. The $f_i(t_k)$ represents the frequency that subsequently varies $t_k$, meaning that the spectral line has unpredictable fluctuations. The $\varphi_k \in [0, 2\pi]$ is the initial phase, and $n(t_k)$ represents the sampling point of $\alpha$-stable distribution noise in $t_k$-th.

The underwater acoustic channel contains Gaussian and non-Gaussian impulse noises. Before establishing the dataset, the modeling and statistical analysis of the measured marine environmental noise were performed. We first modeled three typical marine ambient noises with normal and $\alpha$-stable distributions. Figure 4 shows that the $\alpha$-stable distribution is approximate to the marine ambient noise, particularly at high pulse intensities, which conforms with the results reported in [14]. Subsequently, the characteristic function method [35] was applied to estimate the $\alpha$-stable distribution parameters and statistically acquire the parameter-distribution regularities. Figure 5 presents the statistical conclusions for the four parameters estimated by the $\alpha$-stable distribution. The $\alpha$ is distributed between 1.7 and 2.0, indicating that the analyzed marine-ambient-noise data contain weak pulse characteristics. The $\beta$ is distributed at approximately 0, indicating that the S$\alpha$S distribution can model the noise. The $\gamma$ values are relatively low, ranging from 0 to 0.01. The data amplitudes are relatively concentrated, which is consistent with the weak pulse characteristics. The $\delta$ is distributed at approximately 0, indicating that the measured noise data are concentrated around the zero value. Therefore, the simulation dataset was synthesized according to the distribution regularities of the parameters above and Equation (30).

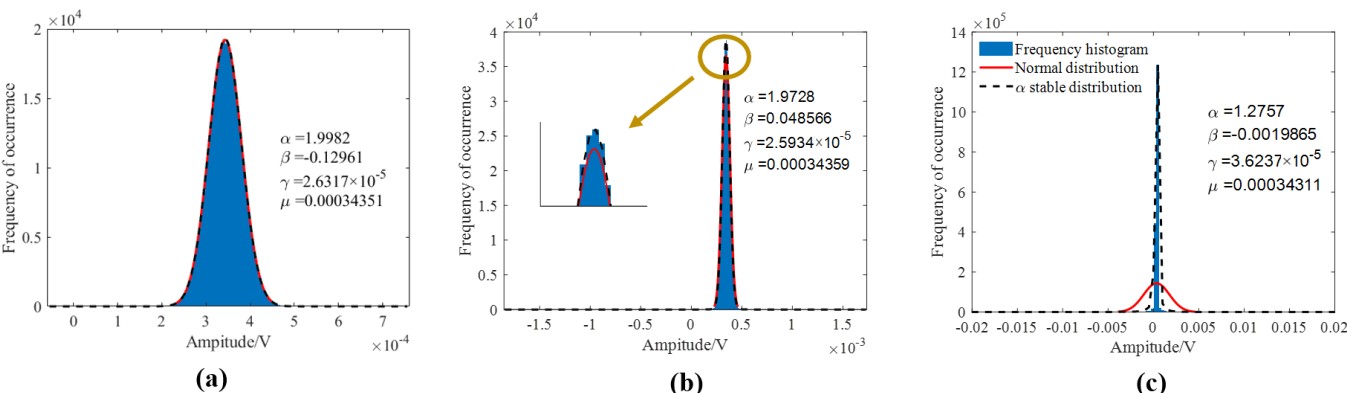

**Figure 4.** Comparison of the modeling of the normal distribution and $\alpha$-stable distribution under various disturbances. (**a**) In a quiet environment; (**b**) in a ship-interference environment; (**c**) under airgun interference.

In the simulation of the S$\alpha$S distribution noise, $\alpha$ was randomly selected in the range of [1.3, 2], $\beta$ is set to 0, and $\gamma$, $\delta$ were set to 1 and 0, respectively. The fluctuating spectral lines within 100 Hz and MSNR in the range of $[-26, -19]$ dB were considered. The sampling rate $f_s$ was 1000 Hz. Our synthetic dataset contained one to five fluctuating spectral lines, and multiple spectral lines had harmonic relations. The S$\alpha$S distribution noise was added to the time-domain amplitude of sinusoids in the form of Equations (28) and (29) with the MSNR and SNR. Figure 6 presents the $H_1$ lofargrams of multiple sinusoidal signals of different MSNRs and $H_0$ lofargrams. The presence of spectral lines in lofargrams is not perceived through the visual senses below $-22$ dB. For MSNR in the range of $-22$ dB to $-26$ dB, we repeated the Monte Carlo simulation 1200 times to simulate the scenario under various parameters, splitting the dataset into 85% for training and 15% for testing.

Therefore, our training datasets comprised 9600 $H_1$ lofargrams and 6800 $H_0$ lofargrams, while the test set had 1440 $H_1$ lofargrams and 1200 $H_0$ lofargrams.

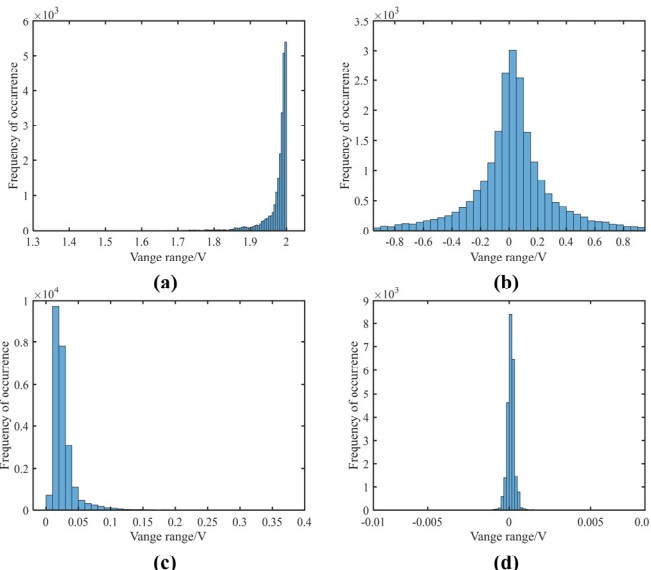

**Figure 5.** Estimation results of $\alpha$-stable distribution parameters. (**a**) $\alpha$-value distribution statistics; (**b**) $\beta$-value distribution statistics; (**c**) $\gamma$-value distribution statistics; (**d**) $\delta$-value distribution statistics.

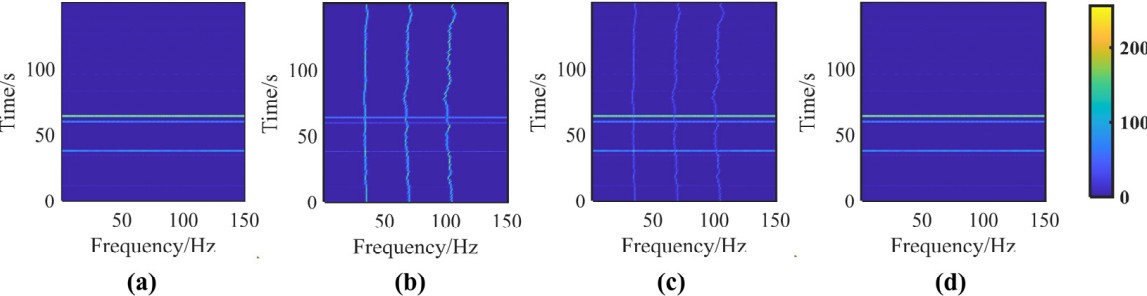

**Figure 6.** Lofargrams: (**a**) only the S$\alpha$S distributed noise lofargram under $H_0$; (**b**) lofargram of the signal at MSNR = $-5$ dB under $H_1$; (**c**) lofargram of the same signal at MSNR = $-15$ dB under $H_1$; (**d**) lofargram of the same signal at MSNR = $-22$ dB under $H_1$.

### 3.2. Evaluation Metrics

The following assessment metrics were utilized to analyze the detection and reconstruction performance.

First, the receiver operating characteristic (ROC) curve was used to evaluate the detection performance. Using Equations (22)–(24), we set various $P_f$ to obtain thresholds in the offline training stage. A serial set of $P_f$ and $P_D$ representing the points of the curve could be obtained, and these points together formed the ROCs.

Second, to evaluate the quality of the reconstruction lofargrams, the mIoU [36] and line-location accuracy (LLA) [37] were employed.

$$\text{mIOU} = \frac{1}{k+1}\sum_{i=0}^{k}\frac{TP}{FN+FP+TP}. \tag{31}$$

where $TP$, $FN$, $FP$, and $TN$ denote the true positives, false negatives, false positives, and true negatives, respectively.

$$\text{LLA} = \frac{1}{\max(|B_1|,|B_2|)}\sum_{(m,n)\in G_1}\frac{1}{1+\lambda\min_{(i,j)\in G_2}(\|[m,n]-[i,j]\|^2)}. \tag{32}$$

where $|B_1|$ and $|B_2|$ denote the accumulation of non-zero elements in the predicted lofargram map $G_1$ and actual lofargram map $G_2$, respectively. The $\|[m,n]-[i,j]\|^2$ indicates the Euclidean distance between the detected spectral lines and the actual spectral lines. We set $\lambda = 1$, as in [37].

### 3.3. Performance Analysis and Discussion

#### 3.3.1. Necessity of AINP

Figures 7 and 8 compare the performances of the AINP under different intensity levels of impulse noise. As shown in Figure 7, heavy SαS noise creates broadband interference in lofargrams. At MSNR = −22 dB, the interference gradually increased as the value decreased. After the preprocessing with the AINP method, the broadband interference in the lofargrams was largely suppressed. However, the spectral-line pixels of the lofargram were still mixed with the low-amplitude impulse-noise pixels and were not visually distinguishable. The following LR-DRNet further processed lofargrams containing a significant amount of low-amplitude impulse-noise pixels. To further indicate the necessity of the AINP in the proposed method, Figure 8 presents a comparison of the performances of LR-DRNet and AINP+LR-DRNet. As shown in Figure 8, when $\alpha = 1.9$, for the cases of −22 and −23 dB, LR-DRNet and AINP+LR-DRNet exhibited comparable performances. As the MSNRs were further reduced to −25 and −26 dB, the performance of AINP+LR-DRNet was better than that of the LR-DRNet. This implies that LR-DRNet has the ability to adapt to weak impulse noise. However, when $\alpha$ decreased to 1.6, the performance of LR-DRNet degraded dynamically. Thus, it can be concluded that the AINP can effectively suppress the broadband interference caused by heavy SαS noise in lofargrams and is necessary for our method to detect and reconstruct weak spectral lines under non-Gaussian impulsive noise.

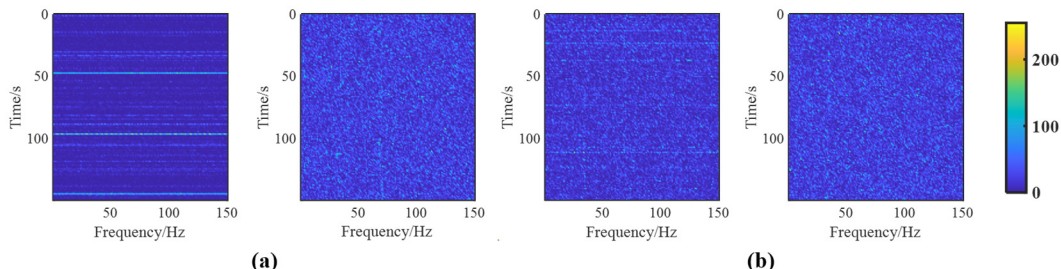

**Figure 7.** Comparison of original and AINP outcomes with different values of $\alpha$ at MSNR = −22 dB. (**a**) $\alpha = 1.6$; (**b**) $\alpha = 1.9$.

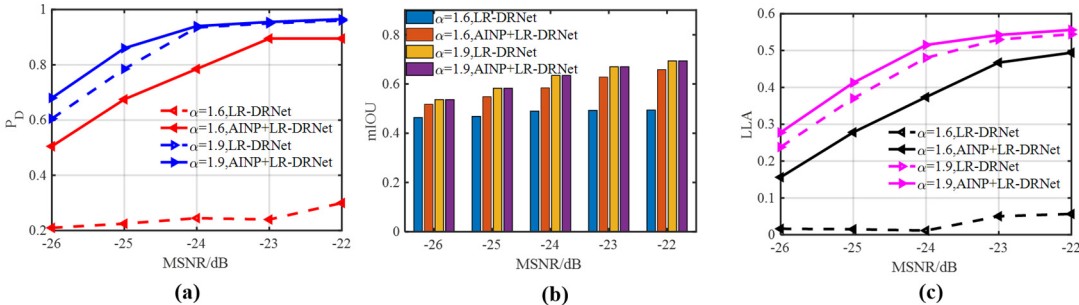

**Figure 8.** Performances of the LR-DRNet and the AINP+LR-DRNet under $\alpha = 1.6$ and $\alpha = 1.9$. (**a**) ROC; (**b**) mIOU; (**c**) LLA.

#### 3.3.2. Network-Structure Analysis

Specific tasks may require suitable network structures. The simulation analyzed the appropriate network structure for spectral-line detection and reconstruction. The two network structures chosen for this analysis were LR-DRNet18 and LR-DRNet34. As shown

in Figure 9, the impact of LR-DRNet depth on performance varies across all MSNRs. Compared with LR-DRNet18, the deeper LR-DRNet exhibited comparable performances in the cases of −22 and −23 dB, and exhibited better performances with −24, −25, and −26 dB, respectively. This shows that increasing the network depth improves network performance. Therefore, the coding layer of the LR-DRNet was set to 34 layers.

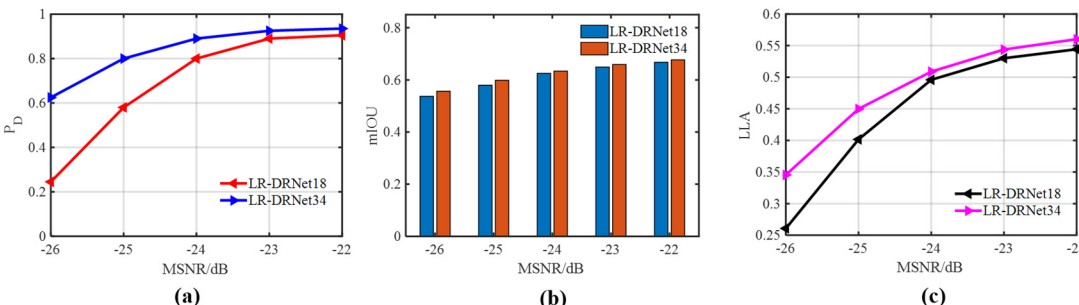

**Figure 9.** Performances of the AINP+LR-DRNet with 18 and 34 depths in different MSNRs. (**a**) ROC; (**b**) mIOU; (**c**) LLA.

Theoretically, the relevance of each feature channel can be automatically determined by the SE structure through learning. This learning of the SE structure determines the significance of each feature channel, which consequently strengthens the desirable features. Therefore, the SE structure needed to be analyzed to determine the performance of the proposed LR-DRNet. As shown in Figure 10, compared with LR-DRNet without SE, LR-DRNet with SE had a significant improvement in detection performance, especially at −25 and −26 dB, along with a slight improvement in reconstruction performance. The parameters of the SE structure participated in the end-to-end network parameter optimization process and optimized the encoder and decoder. Thus, we conclude that the SE structure can significantly enhance detection and reconstruction.

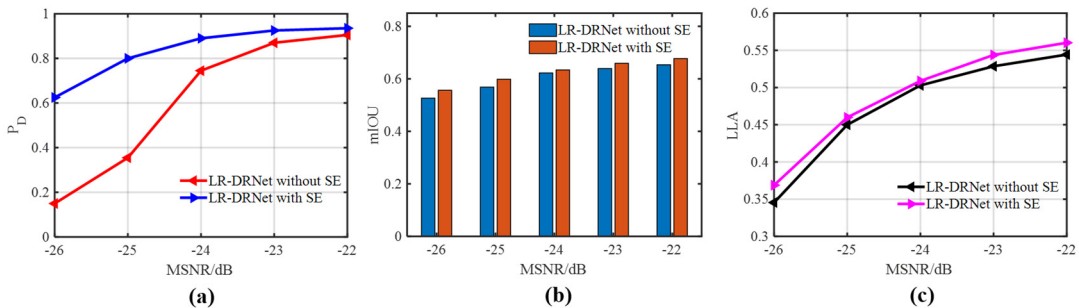

**Figure 10.** Performances of the AINP+LR-DRNet with and without SE in different MSNRs. (**a**) ROC; (**b**) mIOU; (**c**) LLA.

### 3.3.3. Detection and Reconstruction Performance Evaluation

With AINP used as a preprocessor, the outcomes of the proposed AINP+LR-DRNet were compared under various $\alpha$ values and MSNRs. As shown in Figure 11, the performance gradually decreased with decreases in alpha and MSNR, especially at $\alpha = 1.3$ and 1.5. The performances were comparable at high MSNR and $\alpha$ values, presenting more advantages at lower MSNR and $\alpha$ values. At an MSNR of −24 dB and an $\alpha$ of 1.7, the proposed AINP+LR-DRNet still had a $P_D$ of approximately 78%, a mIOU of 0.59, and a LLA of 0.42. In particular, the stronger impulsive noise intensity and lower MSNR affected the feature-extraction ability of the network, encumbering the detection and reconstruction. Nevertheless, the proposed AINP+LR-DRNet is adaptable to low MSNR and strong impulse-noise intensity.

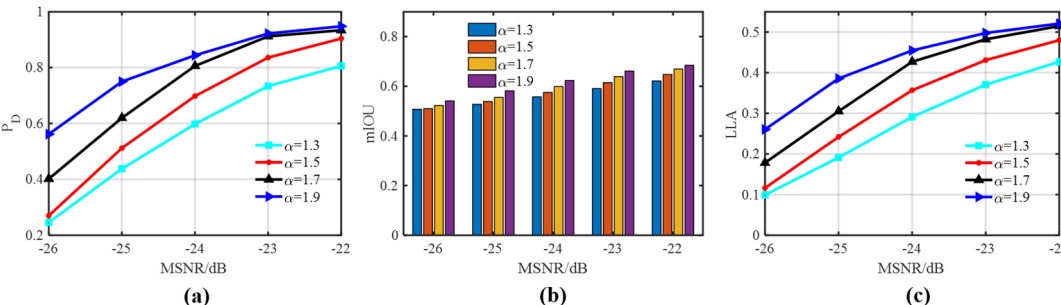

**Figure 11.** Detection and reconstruction with the SαS distribution noise, with α values of 1.3, 1.5, 1.7, and 1.9. (**a**) ROC; (**b**) mIOU; (**c**) LLA.

To verify the feasibility of the proposed LR-DRNet under Gaussian noise, its performance under Gaussian noise was compared with those of other methods. The LR-DRNet34 under a single detection task (LR-DNet34), HMM [14], UNet [24], SegNet [25], ResNet18 [38], ResNet34 [38], and LR-DRNet34 under a single reconstruction task (RNet34) were introduced for performance comparison. To ensure that this comparison was fair, ResNet used the same spectral-line-detection algorithm as LR-DRNet.

Figure 12 compares the detection performances of the proposed LR-DRNet with that of several deep learning methods under Gaussian noise. The proposed LR-DRNet achieved a higher detection rate, particularly at SNR values of −24 dB to −26 dB. Figure 13 presents the differences in the reconstruction performances of the five methods. The reconstruction performance of the proposed LR-DRNet was slightly better than that of RNet34, and better than that of the HMM and other deep-learning methods. In terms of reconstruction, as shown in Figure 14, the proposed LR-DRNet reconstructed weak spectral lines more accurately than the other methods at -25 and −26 dB, while exhibiting comparable performances at −22 and −23 dB. This was consistent with the analysis shown in Figure 13. The excellent detection and reconstruction performance of the proposed LR-DRNet and the superiority of MTL over single-task learning (STL) are illustrated. Thus, the feasibility of the proposed LR-DRNet method under Gaussian noise is illustrated.

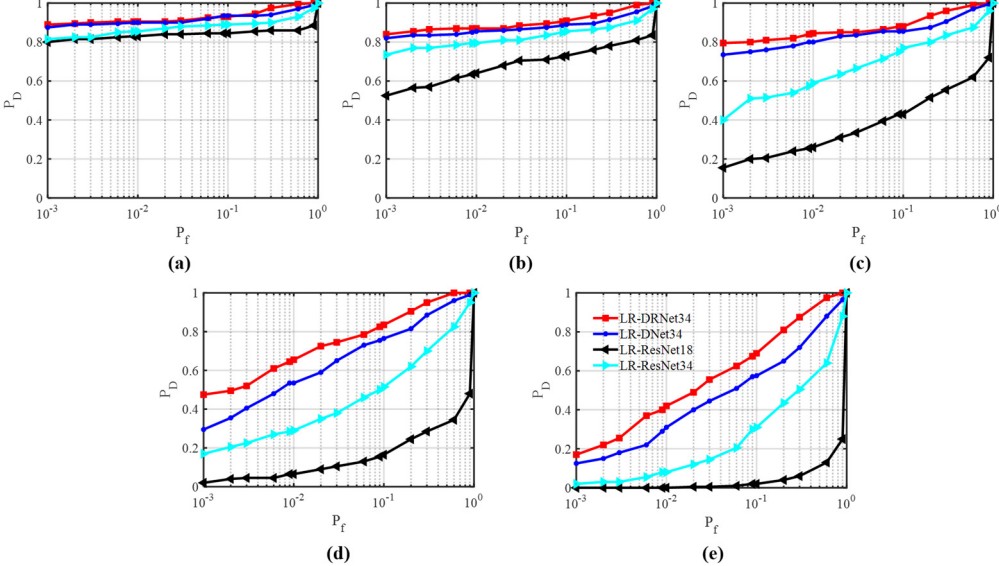

**Figure 12.** Comparison of ROCs of different methods under Gaussian noise in different SNRs. (**a**) SNR = −22 dB; (**b**) SNR = −23 dB; (**c**) SNR = −24 dB; (**d**) SNR = −25 dB; (**e**) SNR = −26 dB.

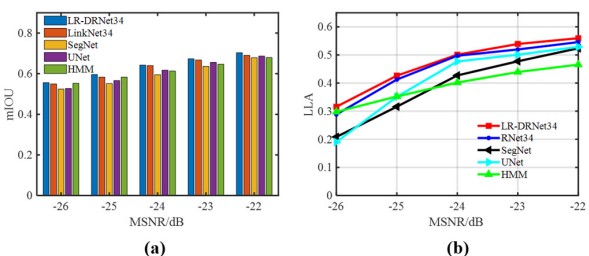

**Figure 13.** Comparison of the reconstruction performance of LR-DRNet, RNet34, SegNet, UNet, and HMM under different SNRs. (**a**) mIOU; (**b**) LLA.

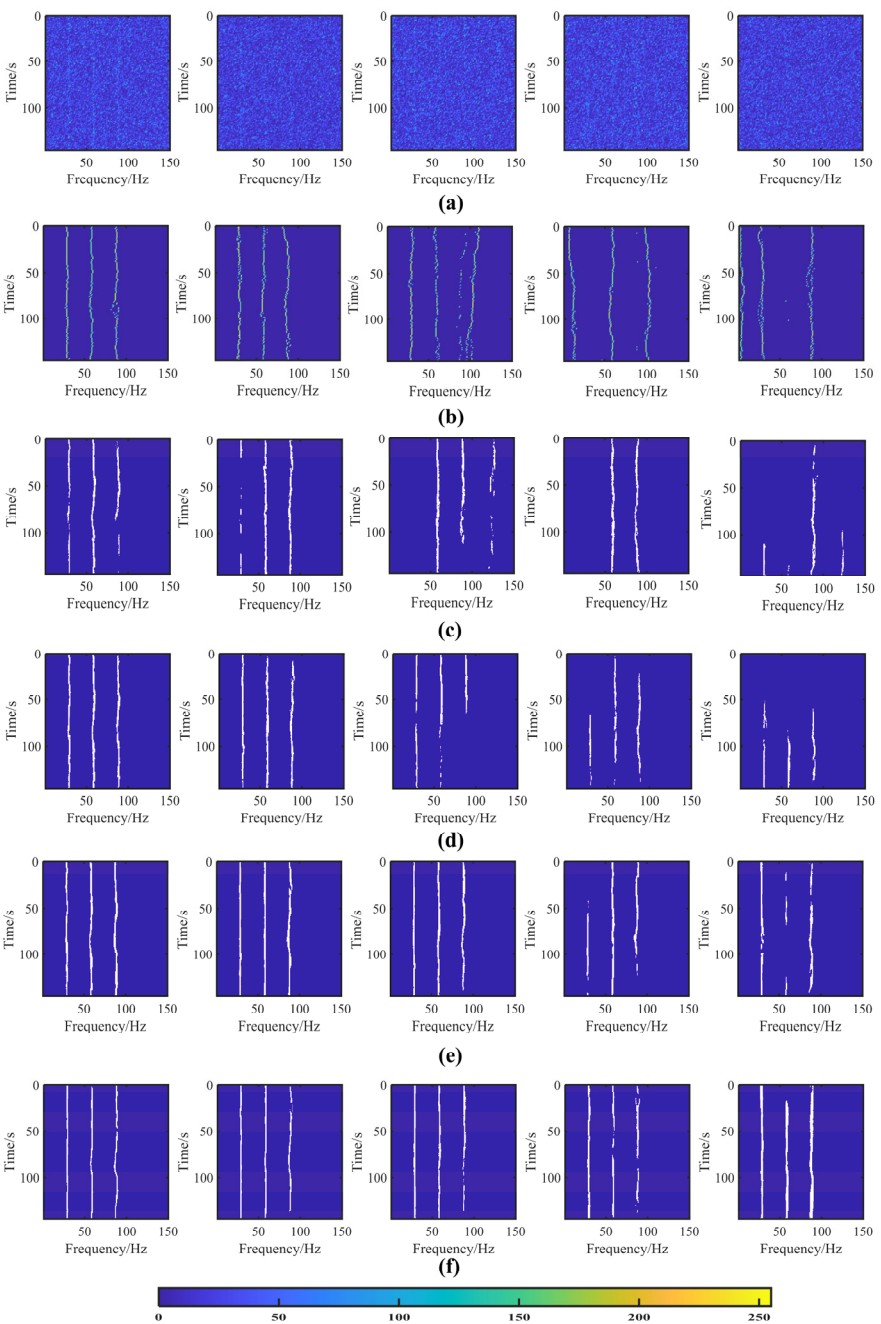

**Figure 14.** Reconstruction of five methods in different SNRs. The original lofargrams with SNR in the range of [−26, −22] dB are shown in (**a**). The same samples were reconstructed by HMM, SegNet, UNet, RNet34, and LR-DRNet34 in different SNRs, as shown in (**b**–**f**).

### 3.3.4. Comparison with Existing Methods

The detection and reconstruction performances of the proposed AINP+LR-DRNet were compared with those of other methods. Deep classification networks, such as ResNet34 [38] and DNet34, and a detector based on a Gaussian function (GF) [39] were introduced for the detection. Semantic segmentation structures, such as UNet [24], Seg-Net [25], RNet34, and HMM [14] were introduced for the reconstruction. In GF, the scale parameter c was set to 2.0, and the impulse intensity $\alpha$ and the dispersion coefficient $\gamma$ were considered in plotting the ROC curve. The number of search times of the spectral line was set to four in the HMM. For a fair comparison, AINP and the algorithm in Section 2.1.4 were used for all the comparison algorithms, except GF.

First, we compared the detection performances of various methods with that of the proposed AINP+LR-DRNet. Figure 15 presents the ROCs of the four methods for MSNR from $-22$ dB to $-26$ dB. The AINP+LR-ResNet34, AINP+LR-DNet34, and the proposed AINP+LR-DRNet exhibited discrepancies, particularly at low MSNR values. Furthermore, the GF and the proposed AINP+LR-DRNet at the same $P_f$ were compared. The GF detector filtered out impulse noise with large amplitudes via a nonlinear transformation, which suggests it had the worst performance. The superiority of the proposed AINP+LR-DRNet in detection is attributable to its specially designed network, which matches the spectral-line-detection algorithm, which is highly capable of feature extraction. The structures of other advanced networks and the disadvantages of the features of the traditional detection algorithm at a low MSNR may hinder detection.

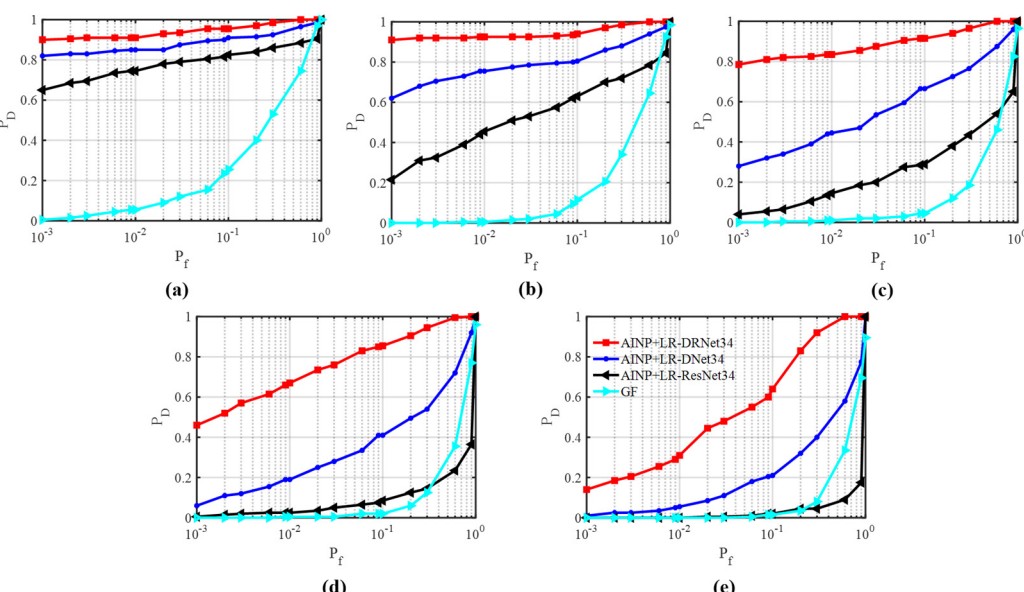

**Figure 15.** Comparison of ROCs of four methods under the S$\alpha$S distribution noise for different MSNRs. (**a**) MSNR = $-22$ dB, (**b**) MSNR = $-23$ dB, (**c**) MSNR = $-24$ dB, (**d**) MSNR = $-25$ dB, and (**e**) MSNR = $-26$ dB.

Subsequently, we compared the different reconstruction methods. The SegNet, UNet, LinkNet34, and the proposed AINP+LR-DRNet are encoding and decoding networks, which segment features with different scales and complex boundaries by extracting the features of the encoding layer and reconstructing the decoding layer. As indicated in Tables 1 and 2, the proposed AINP+LR-DRNet outperformed the other methods by a considerable margin. Specifically, the mIOU and LLA of the AINP+HMM among the five MSNRs ranged from 0.4841 to 0.5566 and 0.2336 to 0.3985, respectively. The mIOU and LLA of AINP+SegNet and AINP+UNet in the five MSNRs were approximately 0.4917 to 0.6719 and 0.0246 to 0.5757, respectively. Accordingly, AINP+RNet34 was superior to the previous three methods, ranging from 0.5305 to 0.6881 and 0.2118 to 0.5859, respectively;

however, the proposed AINP+LR-DRNet achieved impressive performances, ranging from 0.5387 to 0.6932 and from 0.2777 to 0.5950, respectively. Figure 16 presents the lofargram reconstruction of the five methods. The lofargrams reconstructed by the AINP+HMM appeared as false spectral-line pixels after −23 dB, and the line profile became cluttered. The AINP+UNet and AINP+SegNet still worked at −22 dB, but the spectral line broke at varying degrees after −23 dB, and their integrity was reduced. The proposed AINP+LR-DRNet had a prominent spectral-line profile and a higher integrity at −22, −23, and −24 dB, respectively. At −25 and −26 dB, the spectral line could not be reconstructed in some positions because of the excessive background noise. The unique design of the network structure is more suited to reconstruction than those of other segmentation structures.

**Table 1.** mIOU values of different methods for different MSNRs.

| Methods | MSNR/dB | | | | |
|---|---|---|---|---|---|
| | **−22** | **−23** | **−24** | **−25** | **−26** |
| AINP+HMM | 0.5566 | 0.5383 | 0.5236 | 0.5047 | 0.4841 |
| AINP+SegNet | 0.6584 | 0.5909 | 0.5301 | 0.5027 | 0.4917 |
| AINP+UNet | 0.6719 | 0.6205 | 0.5660 | 0.5205 | 0.4991 |
| AINP+RNet34 | 0.6881 | 0.6655 | 0.6300 | 0.5833 | 0.5305 |
| AINP+LR-DRNet34 | 0.6932 | 0.6688 | 0.6316 | 0.5905 | 0.5387 |

**Table 2.** LLA values of different methods for different MSNRs.

| Methods | MSNR/dB | | | | |
|---|---|---|---|---|---|
| | **−22** | **−23** | **−24** | **−25** | **−26** |
| AINP+HMM | 0.3985 | 0.3599 | 0.3277 | 0.2840 | 0.2336 |
| AINP+SegNet | 0.5527 | 0.4182 | 0.1916 | 0.0734 | 0.0246 |
| AINP+UNet | 0.5757 | 0.5169 | 0.3406 | 0.1416 | 0.0499 |
| AINP+RNet34 | 0.5859 | 0.5774 | 0.5221 | 0.4328 | 0.2118 |
| AINP+LR-DRNet34 | 0.5950 | 0.5783 | 0.5373 | 0.4424 | 0.2777 |

Finally, we trained AINP+DNet34, AINP+RNet34, and the proposed AINP+LR-DRNet model separately to examine the validity of the MSL. As displayed in Figures 15 and 16, the proposed AINP+LR-DRNet improved the detection and reconstruction performances after utilizing an adaptive weighted loss function based on dual classification. Owing to the multi-task loss function, detection and reconstruction tasks complement each other by sharing valuable information.

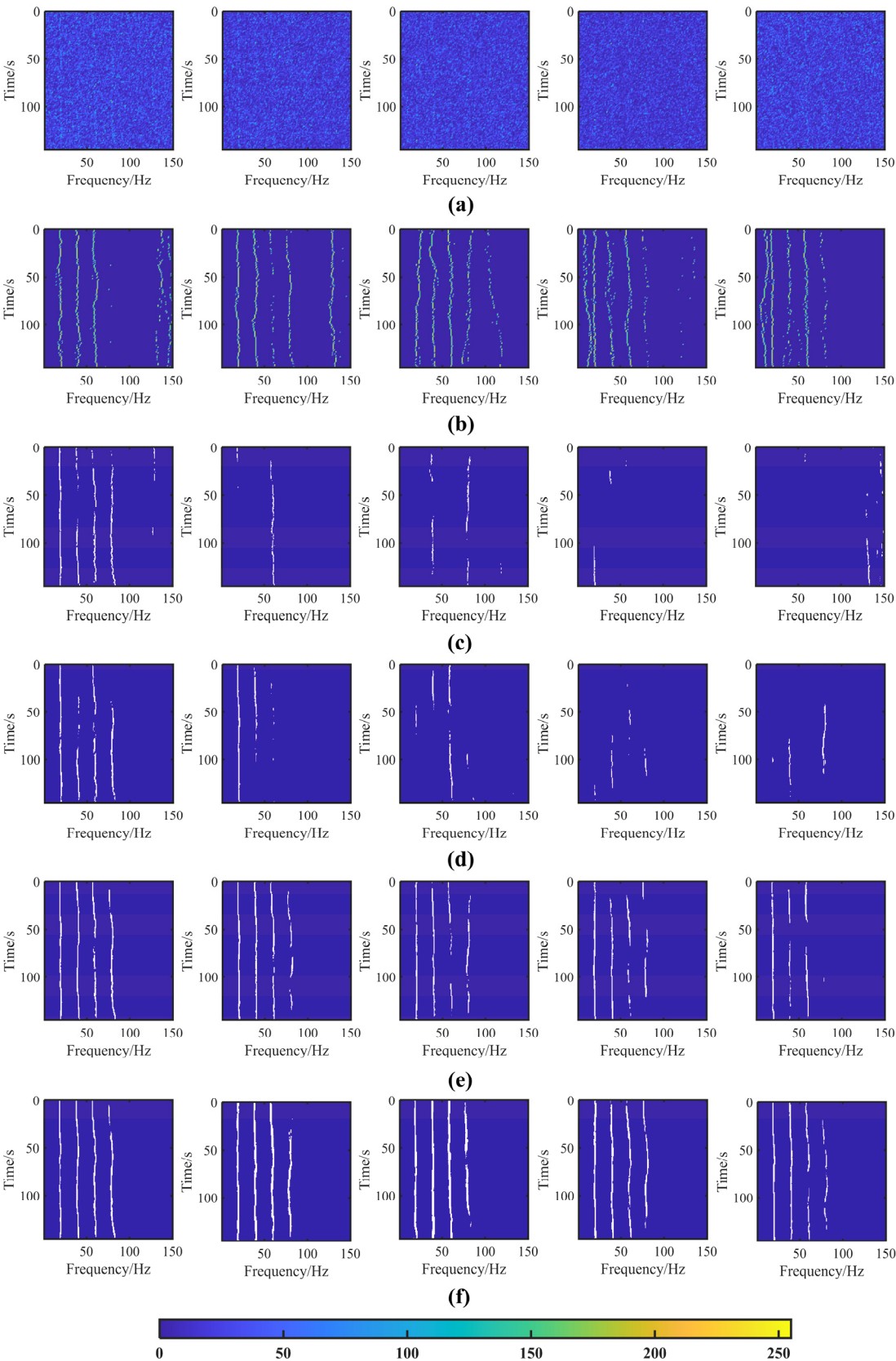

**Figure 16.** Reconstruction results of different methods in different MSNRs. The original lofargrams as shown in (**a**). Reconstructed by AINP+HMM, AINP+SegNet, AINP+UNet, AINP+RNet34, and the proposed AINP+LR-DRNet with MSNR in the range of [−26, −22] dB, as shown in (**b**–**f**).

## 4. Experimental Data Analysis

The detection and reconstruction of the proposed AINP+LR-DRNet in Gaussian/non-Gaussian impulsive noise were verified by the aforementioned simulation analysis. At this point, the weights of the pre-trained model in the simulation were fine-tuned using experimental data. The ability of the proposed AINP+LR-DRNet to detect and reconstruct single and multiple weak spectral lines was analyzed by employing two different experimental datasets, and the performances were compared with those of other methods.

### 4.1. Reconstruction of Weak Single Spectral Line from Strong Background Noise

The data for single-spectral-line detection and reconstruction were received from an experiment conducted in the South China Sea in July 2021. A vertical line array (VLA) composed of 32 hydrophones with an interval of 2 m was employed at a depth of 275–337 m. The sampling rate of the acoustic collector was 10 kHz. During the experiment, the sound source transmitted a single-frequency signal of 71 Hz and was towed 1.5 to 11 km away from the receiving array at a depth of approximately 20 m. Figure 17 displays the hydrophone arrays used in our experiment. We intercepted 2k signal and noise samples from VLA-1 to VLA-32, which formed the measured sample set.

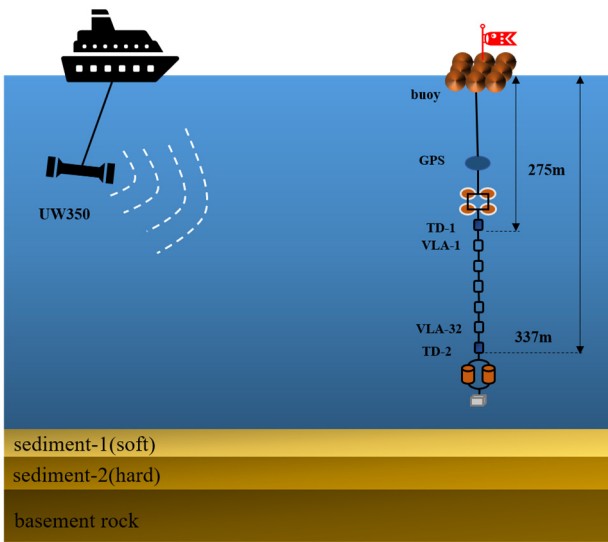

**Figure 17.** Schematic of ship movement and VLA deployment.

In Figure 18a, two unreconstructed lofargrams are displayed in the experimental data, with a relatively weak spectral line. The AINP+HMM, AINP+RNet34, and the proposed AINP+LR-DRNet effectively reconstructed the regions with obvious spectral lines, as shown to the left of Figure 18b–d. As the MSNR was low, the HMM reconstructed some false spectral-line pixels, and AINP+RNet34 reconstructed a few spectral-line pixels. The weak spectral line was reconstructed using the proposed AINP+LR-DRNet, despite the strong background noise in the two cases. Meanwhile, the experimental data show that MTL outperformed STL. Consequently, the proposed AINP+LR-DRNet is suitable for extracting weak single spectral lines from noise-dominated lofargrams.

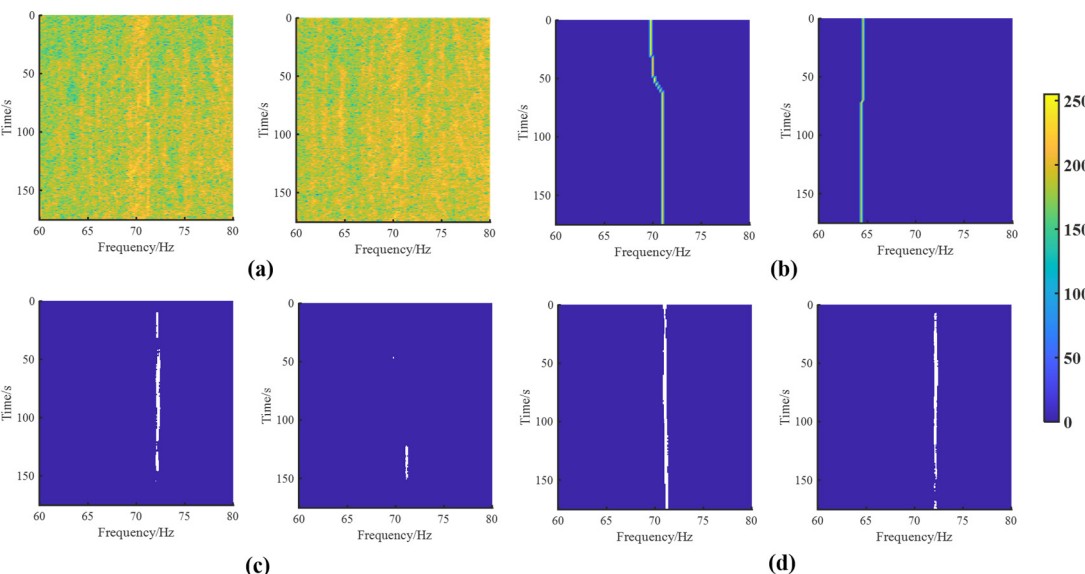

**Figure 18.** Lofargram reconstruction results of experimental data-1 by AINP+HMM, AINP+RNet34, and the proposed AINP+LR-DRNet, shown in (**b**–**d**), respectively. (**a**) Original lofargrams, which were not reconstructed.

### 4.2. Weak Multiple-Spectral-Line Reconstruction against Strong Interference Background

Another experiment conducted in the South China Sea in September 2021 was used to detect and reconstruct multiple spectral lines. An ocean-bottom seismometer (OBS) was deployed every 5 km, with a line length of more than 100 km. The entire seabed was initially relatively flat, and it gradually became inclined near the destination. The sampling rate of the OBS was 100 Hz. As shown in Figure 19, the ship sailed along a straight line at a certain speed for the deployment and recovery of the OBS. Therefore, the OBS can collect ship-noise samples at low SNRs, as well as marine-ambient-noise samples. The test data set was formed with 5000 more signal and noise samples from OBS-1 to OBS-25.

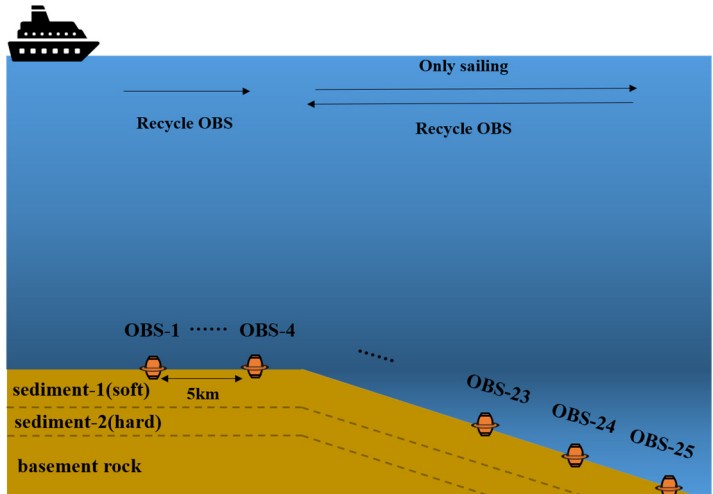

**Figure 19.** Schematic of OBS deployment and recovery in the experiment.

As shown in Figure 20a, the spectral lines of the ship were affected by the strong interference. In one of the cases on the left, the spectral lines at 25 Hz and 33 Hz were blurred on the original lofargram. Furthermore, in another case, the original lofargram did not present a spectral line at 16, 25, or 33 Hz. Figure 20b indicates that AINP+HMM can only reconstruct spectral lines with higher SNR, but becomes ineffective under strong

interference. In addition, more spectral-line pixels were reconstructed using AINP+RNet34, as shown in Figure 20c. The proposed AINP+LR-DRNet is more appropriate for spectral line reconstruction than HMM and RNet34, thereby highlighting the spectral lines and suppressing noise. Hence, the proposed AINP+LR-DRNet is applicable for multiple weak ship spectral line reconstruction under intense interference.

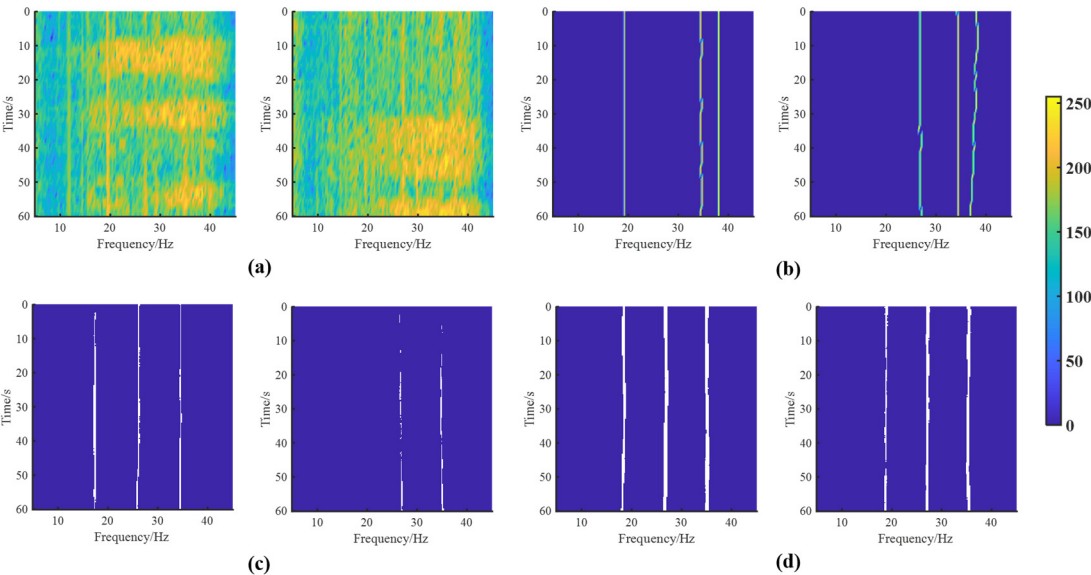

**Figure 20.** Lofargram reconstruction results of experimental data-2 by AINP+HMM, AINP+RNet34, and the proposed AINP+LR-DRNet in (**b–d**), respectively. (**a**) Two different original lofargrams.

*4.3. Detection Performances with Two Real-World Data*

Finally, the detection performances of the GF, AINP+ResNet18, and AINP+DNet34 were compared to evaluate the proposed AINP+LR-DRNet. The $P_f$ was certain for a fixed test set. For a fair comparison, the GF was compared with the detection rate under the false alarm rate obtained by the proposed AINP+LR-DRNet.

As summarized in Table 3, GF displayed the lowest values at low SNR. Compared with GF, the $P_D$ and $P_f$ of AINP+LR-DNet34 were higher. The proposed AINP+LR-DRNet exhibited the highest $P_D$ for the two measured datasets, reaching 94.73% and 94.49%, respectively. Values of $P_f$ of 2.21% and 5.93% were also obtained, which was the best performance of all the methods. This analysis indicates that the proposed AINP+LR-DRNet has the advantage of detection at low SNR under MTL.

**Table 3.** Detection performances on practical data.

| Methods | An Experiment in July 2021 | | An Experiment in September 2021 | |
|---|---|---|---|---|
| | $P_f$ | $P_D$ | $P_f$ | $P_D$ |
| GF | 2.21% | 62.03% | 5.93% | 22.79% |
| AINP+LR-DNet34 | 11.0% | 89.47% | 14.83% | 76.47% |
| AINP+LR-DRNet34 | 2.21% | 94.73% | 5.93% | 94.79% |

## 5. Conclusions

In this study, the joint detection and reconstruction of weak spectral lines under non-Gaussian impulsive noise using DL was investigated. First, with DL, the detection and reconstruction of spectral lines were formulated as a binary classification problem. Subsequently, a framework for weak-line-spectrum detection and reconstruction based on AINP and DRNet was developed. Under the developed framework, a LR-DRNet detection

algorithm was designed, and the lofargrams after the AINP were used as the input of the LR-DRNet. In particular, LR-DRNet was trained by the dual classification adaptive loss to output high detection results and lofargrams with significant spectral lines. Finally, simulated data and real data from the South China Sea were used to verify the performance of AINP+LR-DRNet. The results show that the proposed AINP+LR-DRNet can effectively detect and reconstruct weak spectral lines under non-Gaussian impulsive noise.

In the future, various underwater acoustic signals and marine ambient noises following other distributions will be examined. Furthermore, weak-spectral-line detection based on unsupervised learning will be considered to alleviate the lack of underwater acoustic data and labeling requirements.

**Author Contributions:** Conceptualization, Z.L.; methodology, Z.L.; investigation, Z.L., X.W. and J.G.; data curation, X.W. and J.G.; writing—original draft preparation, Z.L.; writing—review and editing, X.W. and J.G.; visualization, Z.L.; simulations, Z.L.; supervision, X.W. and J.G. All authors have read and agreed to the published version of the manuscript.

**Funding:** We thank the staff of the South China Sea experiment in 2021 for their assistance in providing us with valuable data. This work was supported by the National Natural Science Foundation of China under grant no. 12174078.

**Data Availability Statement:** The data presented in this paper are available upon reasonable request to the corresponding author.

**Conflicts of Interest:** The authors declare no conflict of interest.

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
