# Peer review of "Joint Detection and Reconstruction of Weak Spectral Lines under Non-Gaussian Impulsive Noise with Deep Learning"

_remotesensing, doi:10.3390/rs15133268_

Round 1

Reviewer 1 Report

(1) In Figure 1, "Online detection and reconstruction" uses Online real datasets, how does the model trained in this paper perform on real datasets? What is the expected performance?

(2) Is the meaning of Li and L in Line 122 incorrect, please check.

(3) There are many equations in the text, and there are cases where the same letter represents multiple variables. For example, x is used in equations (2) and (13) to represent input data and labels respectively.

(4) Line 310. The text indicates that the number of "spectral lines" is up to 5. How was this set? Is it randomly set from 1 to 5 or are only 5 lines considered? Also, the examples given in the text are all 3 lines, is this the result of training with 5 lines?

(5) Line 316. What is "various parameters"?

(6) Please give a precise definition of PD, as in equation (30).

(7) In Fig. 12, it appears that the position of the line spectrum is pre-given at the same location. In reality, the positions of the line spectra are random. Can you explain how the random positions of the line spectra are given and how the test data can be examined with random line spectra.

Reviewer 2 Report

In this paper, the authors detect and reconstruct line spectrum under impulsive noise. Whereas, the impulsive noise suppression and line spectrum detection is two individual processes. The novelty is only using an existing neural network for signal reconstruction.

What is the reason of choosing DRNet for signal reconstruction? There is no full name of DRNet. There is no specific reason or improved design of the neural network for underwater acoustic signal detection.

The English writing is hard to understand.

The typing setting is disorder. Some figures and the corresponding titles are not in one page. The typing of math formula looks weird.

Many figures and results are not clearly explained. From Fig. 7, how can the figure have proved that AINP is efficient? There are no comparison algorithms. In Fig. 11, the authors use ResNet for comparison. In the rest figures, there are no comparison with ResNet, but other neural networks. The comparisons are confusing. There is no explanation for figure 13.

Reviewer 3 Report

The comments are written in the attachment.

Round 2

Reviewer 2 Report

The revised manuscript has addressed my comments. The English writing is improved.

1. To enhance the clarity of the employed neural network discussed in Section 2, it is recommended to explain from the perspective of your own system. This can include the neural network architecture (e.g. Figure. 3) and some specific settings (e.g. Eq. 4), et al.

2. Please further proofread the manuscript.
